# Enhancement of Nitric Oxide Bioavailability by Modulation of Cutaneous Nitric Oxide Stores

**DOI:** 10.3390/biomedicines10092124

**Published:** 2022-08-29

**Authors:** Christoph V. Suschek, Dennis Feibel, Maria von Kohout, Christian Opländer

**Affiliations:** 1Department for Orthopedics and Trauma Surgery, Medical Faculty, Heinrich-Heine-University Dusseldorf, 40225 Düsseldorf, Germany; 2Plastic Surgery, Hand Surgery, Burn Center, Cologne-Merheim Hospital, 51109 Cologne, Germany; 3Institute for Research in Operative Medicine (IFOM), University Witten/Herdecke, 51109 Cologne, Germany

**Keywords:** UVA, nitrite, cold atmospheric plasma, nitric oxide donor, skin, wound healing, microcirculation

## Abstract

The generation of nitric oxide (NO) in the skin plays a critical role in wound healing and the response to several stimuli, such as UV exposure, heat, infection, and inflammation. Furthermore, in the human body, NO is involved in vascular homeostasis and the regulation of blood pressure. Physiologically, a family of enzymes termed nitric oxide synthases (NOS) generates NO. In addition, there are many methods of non-enzymatic/NOS-independent NO generation, e.g., the reduction of NO derivates (NODs) such as nitrite, nitrate, and nitrosylated proteins under certain conditions. The skin is the largest and heaviest human organ and contains a comparatively high concentration of these NODs; therefore, it represents a promising target for many therapeutic strategies for NO-dependent pathological conditions. In this review, we give an overview of how the cutaneous NOD stores can be targeted and modulated, leading to a further accumulation of NO-related compounds and/or the local and systemic release of bioactive NO, and eventually, NO-related physiological effects with a potential therapeutical use for diseases such as hypertension, disturbed microcirculation, impaired wound healing, and skin infections.

## 1. Nitric Oxide

Chemically, nitric oxide (NO) is an inorganic gas and can be dissolved in water up to concentrations of 2 mM [1]. In organisms, NO formation is catalyzed by the NO synthases (NOS), which synthesize NO directly from NADPH and L-arginine [2,3]. NO is a biological signal and effector molecule and has a large number of physiological as well as pathophysiological functions in an organism, which depend on concentration, the release profile, and the biological environment, among other things. As the smallest bioactive molecule produced by mammalian cells with lipophilic properties, NO is highly diffusible and can easily cross tissues and cell membranes [1,4]. 

NO controls pivotal physiological functions such as neurotransmission and vascular tone by activation of the soluble guanylyl cyclase [5,6], known to be the primary physiological effector for NO, and also by modulating gene transcription and mRNA translation [7,8]. 

## 2. Enzymatic Nitric Oxide Generation by Nitric Oxide Synthases

No only has a half-life of five seconds, which is why it has to be constantly regenerated. In humans, three isoforms of NO synthase, which are encoded by different genes, are known. These include the two constitutively expressed NOS isoenzymes, which are referred to as nNOS and eNOS, because of their initial discovery in neuronal cells and endothelial cells [9]. However, the expressions of the nNOS and the eNOS are not limited to the two cell types. The eNOS is expressed in many other cell types such as fibroblasts, osteoblasts, or hepatocytes, and the expression of the nNOS is not limited to neurons and is also expressed in skin keratinocytes, among other cells. It is characteristic of the constitutively expressed NO synthases that after agonist activation, in a pulsatile manner, generate small amounts of NO in the pM–nM range over a relatively short period of time [2,9]. For example, the NO generated by eNOS in the endothelial cell lining, the inside of vessels, indirectly causes relaxation of smooth vascular muscle by increasing intracellular cyclic guanosine monophosphate (cGMP) levels, leading to vasodilation and, thus, a reduction in cardiac afterload and blood pressure. This reaction helped to understand how a whole group of drugs worked, including amyl nitrite, nitroprusside, and nitroglycerin: these drugs lead to a direct or indirect release of NO in the body [10,11,12,13]. 

The NO detected in the brain is predominantly the product of the nNOS in neurons. There it takes over the function of a neurotransmitter, whereby it also increases the synthesis of cGMP, among other things. It is also assumed that NO, due to its rapid diffusion, can modulate relatively large areas of the CNS [6,13].

The third NOS is an inducible isoform, the iNOS. After activation by pro-inflammatory mediators (cytokines) and/or bacterial components such as lipopolysaccharides (LPS), iNOS expression is induced, leading to a longer-lasting production of NO in comparatively high physiological concentrations in the µM range [13,14]. For example, the NO generated in high amounts by the iNOS in macrophages or the microglial cells after corresponding activation have fewer signal transductive effects and serve to protect the organism against bacteria, viruses, and helminths due to the radical and toxic character of NO. However, an excessive and, above all, insufficiently regulated high production rate of NO by iNOS can also have side effects such as tissue damage and, for example, profound vasodilatation leading to a dangerous drop in blood pressure, which is observed in the vasculature during septic shock [15,16].

While the role of eNOS- or nNOS-generated NO in human health as a physiological signaling and effector molecule in lower concentrations is not in doubt, many researchers assume that NO at an elevated concentration must have a predominantly negative to pathogenic effect. 

In this context, NO-related diseases can be differentiated by either a lack or excess of NO. For example, NO in the brain regulates many physiological processes that can have an effect on cognitive function and behavior. Moreover, NO promotes angiogenesis and controls brain blood flow, and maintains cell immunity and the survival of neurons. However, an overproduction may result in neurodegeneration [17]. In addition, it was suggested that NO generation may be a major inherited factor of insulin sensitivity and that a diet-induced oxidative scavenging of NO is the first hit toward insulin resistance [18]. In the vasculature, NO from eNOS and nNOS that is also present around arterioles controls the vascular tone and blood flow. Moreover, a steady NO production is essential for leukocyte adhesion and platelet aggregation. Aberrations in vascular NO production can result in endothelial dysfunction, which is associated with several cardiovascular disorders, such as hypertension and angiogenesis-associated disorders (for review see [19]). Here, higher levels of NO generated by iNOS induced by chronic or acute inflammatory processes promote atherosclerosis directly or by the generation of NO metabolites such as peroxynitrite [20]. 

Thus, improving or protecting constitutive nitric oxide production in the vasculature may avoid the development of vascular diseases, whereas the inhibition of excessive NO by iNOS could also represent a therapeutic target [21].

## 3. Nitric Oxide and Skin

In the skin, too, NO-related diseases are caused by either a lack or excess of NO. In low concentrations, NO is a signaling molecule with regulatory and homeostatic functions, such as melanogenesis, vasodilation, and protection against environmental challenges [22]. The eNOS activity of endothelial cells in the skin vessels generates small pulses of NO, resulting in a basal level of vascular smooth muscle relaxation [23]. Thus, the inhibition of eNOS impairs local skin circulation, demonstrating the involvement of NO in maintaining resting cutaneous blood flow [24]. A local NO deficiency in skin contributes to vasospasms, which take part in the vasoconstrictive processes of Raynaud’s disease [23]. 

NO can control pathogen growth in skin infections caused by epidermotropic viruses and many different species of bacteria, protozoa, helminths, and fungi that are often susceptible to NO. For example, NO at high concentrations produced by macrophages via iNOS is able to eliminate intracellular pathogens such as the *Leishmania* species and *Mycobacterium leprae*. It was presumed that this type of NO-induced antimicrobial efficacy would be restricted to macrophages. However, it has become obvious that many cell types, in tissues with immunological barrier functions in particular, use the iNOS-derived NO for defense, contributing to innate immunity [25,26,27]. 

The dark side of a high-output NO generation by iNOS may be seen in the pathogenesis of immune-mediated inflammatory skin diseases such as cutaneous lupus erythematosus, psoriasis, and possibly allergic skin lesions [28,29,30].

NO is a key molecule in dermal wound healing and tissue regeneration and here, too, the NO concentration can determine its function [31]. In primary cell cultures of human keratinocytes, low NO concentrations increased cell proliferation, whereas differentiation was blocked. Using higher NO concentrations, keratinocyte proliferation was inhibited and keratinocyte differentiation was induced. Analogous experiments with human dermal fibroblasts showed a decrease in proliferation correlated with increasing NO concentrations [32]. Independently of any MMP and TIMP action, NO exerts direct regulating properties on collagen metabolism [33,34]; thus, the inhibition of enzymatic NO synthesis causes a significant decrease in collagen synthesis and a delayed wound contraction in rats. In contrast, in vivo transfection of healthy rats with iNOS-cDNA resulted in enhanced collagen accumulation in cutaneous wounds due to increased NO generation [35].

## 4. Nitric Oxide and Nitric Oxide Derivates

Nitric oxide, as a free radical, reacts or binds with a wide range of biomolecules in humans such as proteins at heme, sulfhydryl sites, and cysteine residues, thereby regulating crucial cell functions. Many different bioactive NO-related compounds are generated in the process, such as S-nitrosylated proteins (RSNO), N-nitrosamine, and metal nitrosyls (RNNO), whereas the sum of RSNOs and RNNOs is called RNXOs. A major part of NO is known to be oxidized to nitrite (NO_2_^−^) and nitrate (NO_3_^−^) [36]. 

The biological catalysts of NO oxidation in an organism are not fully elucidated. Oxyhemoglobin and oxymyoglobin as catalysts are restricted to muscle tissues and blood lumen. Intracellularly, possible catalysts are heme-containing peroxidases (e.g., myeloperoxidase, eosinophil peroxidase, lactoperoxidase), which exert intrinsic NO oxidase activity resulting in the formation of nitrite but not nitrate [37,38]. Such NO oxidation activities are suggested to serve as a catalytic sink for NO in areas of inflammatory processes [37].

The reaction products of NO—(nitrite, nitrate, and RXNO) we call nitric oxide derivates (NODs), but they have many other names, for example NO-related species, NO metabolites, NO-related compounds, or NO derivatives, dependent on the authors. Some of these NODs, in particular nitrite and RSNO, are known to exert NO bioactivity under certain conditions, for example hypoxia, acidosis, or UVA exposure, and contribute to the global NO bioavailability [39,40].

Nitrite has been used for millennia to preserve meat (for review, see [41]). Here, the reduction of nitrite to NO, possibly via S-nitrosothiol formation [42], forms iron-nitrosyl, which gives cured meat a distinctive red color and protection against oxidation and bacterial contamination (for review, see [43]). 

Thus, nitrite in the body either stems from NO synthases using L-arginine as a substrate, from dietary intake, or from reduction of dietary nitrate by commensal bacteria. The main sources of nitrate and nitrite in our diet are green vegetables, such as lettuce and spinach, and root vegetables, water, and cured meat. In the USA and many European countries, the estimated dietary intake range of nitrate is from 31 to 185 mg/day and of nitrite from 0.7 to 8.7 mg/day [44]. It is known that in vivo nitrite and secondary amines can react to produce carcinogenic nitrosamines. Therefore, stringent regulations were enforced to lower nitrate as well as nitrite concentrations in food and water. However, urinary excretion in human volunteers is about 1 mmol nitrate per day when nitrate intake is strictly excluded, and, therefore, in the same range of urinary nitrate levels provided by food [45]. Thus, it is assumed that the amount of nitrate synthesized by NO synthases is comparable to the amount of nitrate ingested with diet (for review, see [46]). The impact of dietary nitrite/nitrate intake on human health is a matter of scientific controversy. On the one hand, inorganic nitrates and nitrites are frequently used to avoid bacterial growth in processed meats, the high consumption of which is linked with a greater risk of cancer of the upper gastrointestinal tract [47]. On the other hand, reviews do not show an association between dietary nitrate consumption and cancer risk [48,49]. The predictions that dietary intake of nitrate and/or nitrite may increase the risk of gastric cancer, extrapolated from animal studies, have not been substantiated epidemiologically [44,50]. On the contrary, several studies have observed the beneficial effects of dietary nitrate supplementation on, for example, blood pressure and endothelial function, demonstrating that NO homeostasis can be restored by nitrite and/or nitrate independent from enzymatic NO sources and may represent a further system for endogenous NO production [51,52,53].

However, it is known that nitrite is capable of causing severe methemoglobinemia with a high mortality following the intake of sodium nitrite, which is often misused for self-poisoning [54]. However, nitrite, when administered in a clinical setting for specific diseases, reveals health benefits because most of the published reports identify NO production as the mechanism of action for nitrite applications [52]. 

The beneficial effects of a higher dietary nitrate intake seem to be related to an increase in NO generation, via the reduction to nitrite by oral commensal bacteria and then nitrite further reduction to NO [55,56]. In this nitrate–nitrite–NO pathway, nitrate is absorbed from the stomach and proximal small intestine into the blood stream, whereas a part of it is actively absorbed by the salivary glands resulting in a nitrate accumulation in the saliva. After excretion, the nitrate in salvia is rapidly reduced to nitrite by commensal oral facultative anaerobic bacteria located in the mouth (sublingual), swallowed, and then absorbed in the gut. Here, it enters the systemic circulation or is partly reduced further to NO under the acidic conditions in the stomach, from where it, in turn, enters the circulation, where it is oxidized to nitrite and nitrate [55,57]. As an overall effect, a higher nitrate consumption increases the general levels of available nitrite/NO in blood, body fluids, and in tissues [58]. Vegetarians are at reduced risk of developing hypertension and other cardiovascular diseases. Therefore, it can be speculated that the high nitrate/nitrite content of many consumed vegetables may possibly contribute to these beneficial cardioprotective effects [41,59], in addition to the often-cited antioxidant effects of vegetables.

There are many pathways in the body involving hemoglobin, myoglobin, xanthine oxidoreductase, or the further reduction of nitrite to bioactive NO, which is particularly enhanced during acidosis and hypoxia. Therefore, it is thought that these mechanisms represent a back-up system to ensure NO generation when oxygen-dependent NOS are compromised [51,55]. 

In the next sections, we present the NOD stores in the human body and introduce the non-enzymatic pathways, which generate NO from nitrite, affecting skin physiology in particular and many secondary systemic parameters.

## 5. NOD Content of Tissues and Skin

As products of enzymatic NO synthesis, nitrite and nitrate are widely distributed in the human body. However, their distribution varies, so that their concentrations in different body fluids can differ significantly [60]. In body fluids such as urine and saliva, nitrate concentrations are found in up to triple-digit micromolar concentrations, whereas in blood plasma, gastric juice, or milk only single- to double-digit micromolar nitrate concentrations are found. Nitrite levels in the human body are generally lower than nitrate levels. Without a bacterial urinary tract infection, no nitrite is found in the urine under physiological conditions, while concentrations of up to 200 µM and higher are observed in the saliva. The concentrations in blood plasma and milk are also very low and often below the detection limit. In the stomach, the nitrite concentrations in the gastric juice can be very variable and can be significantly increased in people with gastric diseases, with up to 200 µM present [61]. Weller et al. were also able to detect relevant amounts of nitrate and nitrite in human sweat [62]. Since it is easier to take body fluids and measure them quickly, there are fewer studies that have examined the concentrations of nitrite and nitrate in different tissues. A study of Nyakayiru et al. showed that, in humans, the content of skeletal muscle nitrate is clearly higher than in blood plasma. However, in this study, the nitrite concentration in skeletal muscle remained below the detection limit [63].

In human skin homogenizates (epidermis + dermis) we showed that the concentration of nitrate was around 6-fold, of nitrite 25-fold, and of RSNO/RNNO up to 40-fold higher than in blood plasma [64]. These results are consistent with another study by Mowbray et al. using another experimental set-up. Interestingly, they found that most of the nitrate and nitrite is found in the epidermis and cornea, and only a small portion of it in the underlying dermis [65]. In addition, they further described a strong inter-individual variation in the concentration of NO-related products in the blood plasma, saliva, sweat, superficial vascular dermis, and epidermis. They also demonstrated that the concentration of NODs found in the blood plasma strongly correlated with the NOD concentration in the superficial dermis or sweat. It was suggested that the majority of nitrite found in tissues may have originated from exogenous dietary intake of nitrite and nitrate instead of from endogenous sources, which may cause great inter-individual variation in tissue nitrite levels, depending on individual nitrate and nitrite intake [41]. In contrast, nitrite blood plasma concentrations are more stable, suggesting the existence of regulatory mechanisms in the blood [66].

In conclusion, normal human skin of healthy volunteers can contain NODs such as nitrite and RSNO in many-fold higher concentrations than in blood plasma. Why and how NODs accumulate in the skin although there is fast renal elimination is a matter of much speculation. There are studies that provide direct evidence that the existing dermal NODs are jointly responsible for NOS-independent NO generation and can thus exert local and systemic NO-specific effects, which we introduce in the following sections.

## 6. NO Generation by Decomposition of Dermal Nitric Oxide Derivates

### 6.1. Acid-Induced Nitrite Decomposition of Nitrite in Sweat

There are many possible ways to generate NO in skin and on the skin surface besides the NOS-dependent NO synthesis. Analogously oral bacteria and also the skin commensal bacteria can synthesize the nitrate reductase enzyme, which reduces nitrate of sweat to nitrite. Owing to the acidic nature of sweat, nitrite is reduced further to NO, which in turn can evaporate in ambient air and also easily cross the epidermal barrier, thus entering the skin tissue and reaching skin cells, and also possibly the superficial blood vessels and the blood circulation [62]. In addition, it was postulated that ammonia-oxidizing bacteria may contribute to superficial dermal nitrite concentration via oxidation of ammonia to nitrite [67]. In previous studies, we found that NO generation by acidification of low concentrations of nitrite (10 µM) depends on the pH value and can be enhanced in the presence of antioxidants such as vitamin C (see Equations (1)–(4))
NO_2_^−^ + H^+^ ⇆ HNO_2_(1)
2 HNO_2_ ⇆ N_2_O_3_ + H_2_O(2)
N_2_O_3_ ⇆ NO + NO_2_^•^(3)
HNO_2_ + Asc → 2 NO + DHAsc + 2 H_2_O (4)

This enhancement is more pronounced under lower pH values (pH 2–4), whereas under normal skin pH values the presence of vitamin C does not play a big role. However, the antioxidant-assisted NO generation can be boosted manifold by copper ions at the pH value of 5.5 and under normoxia conditions [68]. Technically, when using higher nitrite concentrations, the underlying mechanism of antioxidant/copper-assisted NO generation at normal skin pH values can be used as an NO donor system with therapeutic potential, e.g., for increasing dermal microcirculation in critically perfused flaps in plastic surgery [69,70].

There are reports confirming the presence of relevant amounts of water-soluble vitamins, e.g., vitamin C in sweat [71]. Interestingly, these concentrations are in the same range as found in blood plasma [72]. In addition, copper can be found in sweat and, although there was no correlation between serum copper and sweat copper, there were hints that a high dietary intake of copper resulted in larger excreted amounts of copper in sweat [73].

In conclusion, besides nitrate/nitrite concentration of sweat, bacterial nitrate reductase activity, and the sweat pH value, it is very probable that further individual factors (such as the presence of copper ions and antioxidants) have a strong impact on acid-induced nitrite decomposition in sweat and consequent NO generation, making in vivo investigations more difficult.

### 6.2. UV-Induced Photolysis of NOD Stores in Skin 

As early as 1961 the term photorelaxation was introduced by Nobel Prize winner R.F. Furchgott to describe the relaxation of rings of rabbit aorta under light (<450 nm) and UV radiation [74], potentiated in the presence of nitrite [75]. Nitrite can undergo photodecomposition induced by UVA irradiation, resulting eventually in the generation of bioactive NO [76]. Therefore, it is obvious that dermal nitrite can be targeted by UVA, as demonstrated by a study by Paunel et al. Here, UVA exposure of skin specimens causes an enzyme-independent high-output NO generation above the skin surface and within the skin, which not only correlates with the nitrite and RSNO concentrations in the skin but also with concentrations of free and protein-bound thiols, which may serve as antioxidants. Further experiments show that the UVA/nitrite-induced biological effects on keratinocyte differentiation and proliferation could be enhanced in the presence of antioxidants such as vitamin C and glutathione (GSH) [64]. In additional studies, we found that the NO formation from UVA-induced decomposition could be enhanced manifold by antioxidants such as Trolox (water-soluble vitamin E derivate), GSH, and vitamin C [77].

In healthy volunteers, an increase in RXNO and nitrite concentration in the blood plasma could be observed at an interval of 15 to 45 min after a whole-body UVA irradiation, indicating a possible UVA-induced mobilization of nitrite-derived NO from skin tissue into the blood plasma [78]. These effects were accompanied by and correlated with a significant drop in systemic blood pressure, raising the question whether UVA could be good for the heart [79,80]. These results were supported by Mowbray et al., showing in healthy volunteers, via microdialysis, a likewise increase in NO-related products by UVA irradiation [65]. In addition, a study by Liu et al. using confocal fluorescence microscopy and an NO imaging probe on human skin samples revealed that UVA-induced NO release occurs in a dose-dependent manner, with the majority of the light-sensitive NO pool in the upper epidermis, independent of NOS activity. In addition, here, UVA lowered blood pressure independent of NOS [81]. 

It can be assumed that UVA also has an effect on photolabile NODs such as nitrite and other RXNO in sweat, which in turn is produced more under warm and sunny conditions. Our in vitro experiments (data not published, see Appendix A) show that buffers (pH 5.5) containing low concentrations of nitrite release NO upon UVA exposure. As shown in Figure 1B,C, this NO release can be boosted by the addition of an antioxidant, e.g., vitamin C. We observed in vivo that under UVA exposure, the increase in NO evaporation from human skin increased manifold. However, the addition of topical vitamin C could only slightly enhance the NO amount, indicating that the process of UVA-induced decomposition on the skin surface and upper skin layers was already using naturally occurring antioxidants (sweat) in sufficient concentrations (see Figure 1D,E). However, washing of the skin prior to UVA exposure halved the NO yield. In a parallel study, using the same experimental set-up, we showed that heat generation was not primarily responsible for the increase in skin NO release, but the used wavelengths for irradiation (Section 6.3) [82]. 

Investigating the NO release action spectrum in human skin, Pelegrino et al. observed that NO can be generated by UVA and also UVB irradiation, which both could trigger the dermal NOD store, mainly composed of nitrite, nitrate, and RSNOs [83].

It is possible that NO released by photolabile NOD, in particular nitrite, may serve as a protection against UV challenge. In human skin cells, even supra-physiological high concentrations of NO protected cells from oxidative stress and UVA-induced apoptosis. However, in other cell systems, the number of apoptotic events, even at physiological concentrations of NO, increased [84]. Nevertheless, skin fibroblasts depleted from intracellular nitrite showed a higher UVA susceptibility and died at lower doses than control fibroblasts or fibroblast cultures supplemented with physiological nitrite concentrations found in the skin [85].

Thus, it seems that skin-derived nitrite and other NODs may play an important role in human skin physiology, as it is postulated for NO itself. However, further experiments showed that nitrite at higher supra-physiological concentrations enhanced UVA-induced cell deaths in skin fibroblasts, probably due to the generation of toxic NO_2_ radicals produced simultaneously by UVA-induced nitrite decomposition [86]. Here, the addition of the antioxidant ascorbate could reverse the UVA/nitrite-induced toxic effects, whereas GSH and Trolox enhanced them. Although NO formation via photodecomposition of nitrite may serve as an effective antioxidant and protector in skin, the simultaneous generation of toxic side products may have adverse and harmful effects. 

Thus, besides the nitrite concentration, here the individual antioxidative capacity of a cell type and the microenvironment during UVA exposure are also crucial for the outcome and should be considered when dealing with UV/A-induced skin effects such as sunburn, erythema, and premature skin ageing. 

In conclusion, analogous to acid-induced nitrite decomposition, the UV-induced photolysis of NODs depends on the wavelength, irradiance, and dose; individual factors such as nitrate/nitrite/antioxidant concentrations and pH of the skin, skin surface, and sweat; as well as skin hygiene, skin type, and others. In addition, possible interactions with light/radiation of other wavelengths (visible light/IR) should not be forgotten, as they are part of the natural sunlight and can have a direct impact on NO release and NOD stores (see the following section). 

### 6.3. VIS/IR-Induced Photolysis of NOD Stores in Skin

Apart from UVA radiation, we demonstrated that blue light at shorter wavelengths is also able to mobilize NO from photolabile NODs in the skin. We found a significant increase in the intradermal levels of free NO caused by blue light irradiation in human skin specimens. Furthermore, blue light induced an emanation of NO from the skin area in healthy volunteers, whereas other wavelengths in the green, red, and infrared spectrum did not have significant effects [82]. 

In a randomized crossover study conducted by Stern et al., 14 healthy male volunteers were irradiated by monochromatic blue using a full-body blue light device, and the circulating nitric oxide species (nitrite/RXNO) in blood plasma were measured. The results showed that 30 min after the end of irradiation, the levels of nitric oxide species increased in circulation. In particular, the levels of RXNOs were significantly elevated by 30–50% [87].

There are reports that IR and red light may have an impact on NO stores in the skin or NO release, possibly via photolysis [88]. However, further investigation showed that enzymatic pathways were dominant in the induction of NO release found in ex vivo human skin homogenates [89]. In experiments using human keratinocytes, an increase in NOS-dependent NO production was observed after infrared low-level laser stimulation [90]. Since the observed increase in NO production was very quick, the authors postulated that the existing NOS activity may be enhanced instead of a de novo NOS protein synthesis, indicating that IR may stabilize NOS, cofactors, and enhance substrate binding ability, possibly by heating. 

## 7. Modulation of Dermal NOD Content and Possible Effects 

### 7.1. NO Donors

Since NO is important for wound healing and shows antibacterial efficacy in many studies, different types of NO donors such as RSNOs (S-nitrosocysteine, S-nitrosoalbumin S-nitrosoglutathione), N-diazeniumdiolates (NONOates), metal nitrosyls, and others were used [91,92]. There are many reports that state exogenous NO donors represent a promising method to promote wound healing by enhancing cell proliferation, collagen deposition, and angiogenic activities improving granulation tissue formation [93].

Acidified nitrite creams were also used, as NO donor systems showed good therapeutic effects on wound healing in mice. However, the outcome was better when cream was applied in the first 4 days after wounding [94]. In a prospective study (8 patients, 15 infected wounds), the possibility of MRSA eradication was also demonstrated (9 of 15 wounds) using the same cream formula [95]. 

A possible clinical application of NO donors is to improve dermal microcirculation and tissue perfusion, which are often pathologic in patients with hypertension, obesity, and diabetes mellitus, but also often critical after skin flap surgery [96,97]. Many studies have proven the effectiveness of NO donors to improve flap survival in experimental models [98]. In humans, we showed that topically applied NO (acidified nitrite/ascorbate) significantly increased vasodilatation and blood flow. These beneficial effects were also observed in a patient with a critically perfuse flap preventing further surgery [70]. 

Nevertheless, in spite of these beneficial results of using NO donors, they are not very common in clinical practice. In cells, NO can cause cell death, induce DNA damage, and disturb mitochondrial functions. High concentrations of NO are toxic, as seen in inflammatory responses where a high-level activity of iNOS results in high NO concentration associated with the destruction of cells and tissues [99,100,101].

Therefore, one major problem involves the therapeutic windows and possible adverse side effects, as reported in some studies [102,103]. Here, a repeated topical application of acidified nitrite exerted pro-inflammatory effects in the skin, and a dose-related increase in itching, pain, and edema were observed in patients with anogenital warts. A study by Mowbray et al. suggested that the observed potent inflammation by acidified nitrite produced by a combination of nitrite and ascorbic acid was secondary to the release of additional mediators (e.g., ascorbyl radicals), whereas using a chemically inert pure NO donor (NO zeolite) providing the same NO release exerted only minimal inflammation [104].

Nonetheless, we demonstrated in vitro using an acidified nitrite/ascorbate/copper system and gas permeable cell culture well bottoms that the NO release profile was crucial for the outcome. Here, directly after mixing and application of liniments, the obtained initial high peaks of NO amounts in the first 200 s correlated better to cell toxicity than the total amount of applied NO in 600 s [69]. Evaluating potential damage effects on human skin in further studies (see Appendix A), we observed an increase in apoptotic events in epidermis and dermis in freshly donated human skin after topical application of slightly acidified nitrite/ascorbate liniments with different nitrite concentrations (see Figure 2). Here, higher nitrite concentrations led to higher NO penetration through the epidermis (Figure 2B). We observed that the rate of apoptosis in the epidermis correlated with the generated NO amount in the liniment (see Figure 2C,E). Furthermore, within the dermis, the number of apoptotic events was seen to generally increase for all tested NO-releasing liniments, but neither showed a good correlation to NO doses nor to the penetrated NO amounts (see Figure 2C,D,F). These results indicate that different cell populations and possible fibroblast subpopulation in the dermis may show different NO sensitivity, which could be interesting for therapeutic approaches to hyperproliferative fibrotic conditions, for example, excessive scaring after burns.

Most studies relating to the use of NO donors for skin pathologies only investigated the direct NO effects on bacteria, dermal microcirculation, and skin cells. However, less studied is the fate of topically applied NO. Apart from the direct local effects, the application of any NO donor should increase the amount of more stable NODs in the treated area via NO reduction and other chemical reactions, e.g., S-nitrosylation. These NODs may accumulate locally and remain in the treated tissue and/or may be distributed systemically. By using 15N-labeleld nitrite we observed that topical application of NO (acidified nitrite/ascorbate) on skin samples and the skin of healthy volunteers led to a transepidermal translocation of NO into the underlying tissue, resulting in a significant increase in nitrite and RSNO in the skin and blood [70]. Depending on the size of the treated skin area (torso), a significant systemic increase in the NOD (nitrite and RXNO) was also found in blood plasma, accompanied by an increased systemic vasodilatation and blood flow as well as a reduced blood pressure [105]. 

In summary, apart from direct effects on wound healing, bacterial contamination, microcirculation, blood pressure, and others, the topical application of NO donors represent an approach to directly increase circulatory NO bioavailability, and also indirectly by increasing the local and systemic NOD stores, which in turn may have longer lasting effects on, for example, skin physiology/bacterial colonization and circulation parameters.

### 7.2. Cold Atmospheric Plasma

Physical plasma is referred to as the fourth state of matter alongside solid, liquid, and gas and has become increasingly interesting in medical research in recent years. Plasma can be generated at room temperature under normal atmosphere (cold atmospheric plasma; CAP); thus, it can also be applied to sensitive surfaces such as human skin and tissue. By supplying energy, for example, by applying a strong electric field, ions, atoms, and especially electrons of a gas are set in motion. Through impact ionization, electrons are accelerated and catapulted out of their orbits and react with other molecules/atoms generating, among other things, radical nitrogen and oxygen species. Within CAP, other components such as UV rays (UVA, UVB, UVC), ions, neutral atoms, and heat are generated, determining the effects of the CAP [106]. There are many types of CAP sources using different approaches to CAP generation. Direct CAP sources use the surface to be treated as a counter electrode, and the active particles are generated directly on this surface. The energy-generating first electrode is covered with ceramic, so that an attenuated discharge occurs, which discharges homogeneously in the form of many small lightning strikes on an uneven surface. It is important that the distance electrode/surface is uniform and not more than 1–2 mm. A device using this kind of CAP generation is called a “dielectric barrier discharge” or DBD for short, and the ambient air is used as the gas to be ionized. 

Indirect plasmas are generated at two identically constructed electrodes and then transported with a carrier gas (e.g., helium or argon) to the surface to be treated [107]. The plasma becomes visible as a narrow gas jet. The treated surface is not in an electric field, which means that mainly uncharged particles are transported [108].

CAP becomes complex through the various factors that influence its composition and, thus, its effects on cells and tissues/body fluids. Among other things, it is characterized by its particle composition, temperature, type of generation, spatial distribution, and strength of the electric field. These factors can be individually designed and, thus, optimized to suit the field of application. 

CAP has been thought to be used for disinfection and sterilization of surfaces, such as infected wounds, and may have potential besides its antibacterial efficacy to promote wound healing [109,110,111]. However, we found that CAP treatment using a commercially available and clinically approved plasmajet obtained good results on dry surfaces but did not lead to the desired significant reduction in the bacterial burden in a wet wound milieu or in biofilms [112].

Investigating a direct CAP source/DBD (see Appendix A), we found that topical CAP treatment induces NOD accumulation in the treated skin area, as shown in Figure 3A–C, accompanied by an acidification of the skin surface (down to pH 2) and NO-like effects, such as increasing dermal microcirculation without significant toxic effects to skin or skin cells (Figure 3D) [113]. 

Similar results were obtained using a commercially available and clinically approved PlasmaDerm device based on DBD, showing increased cutaneous capillary blood flow at the radial forearm of healthy volunteers after CAP treatment [114].

In summary, apart from the supposed antibacterial and wound healing effects, the topical application of CAP and the use of direct CAP sources, in particular, may represent an alternative approach to increasing dermal and perhaps systemic NOD stores and NO bioavailability (for overview see Figure 4).

Showing similar effects as acidified nitrite/NO donors, CAP may also exert beneficial and detrimental effects, depending on the treatment regimen and dose. Although there are many reports of CAP/DBD-induced cell death, we found in our experiments that CAP/DBD did not induce apoptosis in human donor skin but sometimes a slight inflammatory response [113].

### 7.3. Nitrate-Rich Diet

Assuming that permanently higher NOD blood plasma levels are also reflected in higher NOD concentrations in tissues, oral intake of NOD such as nitrate and nitrite should not only increase plasma levels but also replenish dermal NOD stores. 

Since Mowbray et al. demonstrated that NOD concentration in blood plasma is correlated with NOD concentration in superficial dermis or sweat [65], it is probable that this assumption is right and that the dermal NOD stores can be filled by a regular nitrate-rich diet, preferably with high contents of vitamins and antioxidants by eating vegetables, such as spinach and red beetroot. Moreover, because certain bacteria are essential for the enterosalivary nitrate–nitrite–nitric oxide pathway, disturbing the oral bacterial fauna by, for example, antiseptic mouthwash, may have strong negative impact on NO bioavailability and NOD levels mediated by oral nitrate uptake [115].

To what extent the dermal NOD content can be influenced by nutrition and the possible physiological consequences this may have has not yet been sufficiently investigated and needs further research. However, increasing dermal NOD stores via nutrition and support of the oral bacteria seem feasible and delicious approaches. 

### 7.4. UV/VIS Modulation of Dermal NOD

UV radiation and visible light—blue light in particular—are able to target photolabile NOD in the skin, resulting in NO release. This kind of NO mobilization also leads to possibly long-lasting changes in the dermal NOD stores. Paunel et al. described how UVA irradiation of skin induced an increase in the dermal RSNO content, which can be enhanced by incubation with nitrite [64]. On the other hand, a depletion of photolabile NOD can be expected by UV/VIS, which can be an interesting pathway in photo-induced premature skin ageing. To our knowledge, no study has been conducted in this direction yet. However, in a combination, for example, with a topical neutral nitrite application, UV/VIS may be used to mobilize NO from more superficial NODs (nitrite) into deeper skin layers, where it is possible to be stored as bioactive RSNOs and nitrite, increasing the level of NOD stores in skin. On the other hand, NO donors (acidified nitrite/ascorbate) increased the NOD levels in skin, which in turn may result in a higher NO release upon UVA or VIS. Personal observations indicate that the increased NOD levels in the skin induced by NO donors can remain for days, as was found out by chance when we investigated the skin NO release upon UV irradiation. Here, the irradiation of one forearm of a volunteer, which was in contact with an acidified nitrite/ascorbate cream three days before, led to a manifold stronger (~10×) NO release than the other (uncreamed) arm. 

Thus, elevated dermal NOD levels obtained also, for example, by a nitrite/nitrate-rich diet or inflammation, should be affected by UV/VIS, possibly leading to more dermal/circulatory RSNO/RXNO concentrations with consequent NO-induced effects on blood pressure and microcirculation [87]. 

In order to avoid any misunderstandings, of course, it is appropriate at this point to point out the potential dangers of UV exposure. UV radiation can be a harmful and carcinogenic environmental medium, and frequent UV exposure with higher radiation doses that can lead to erythema or even sunburn should be avoided at all costs [116]. 

However, it is also undisputed that sunlight, as the most important physical environmental factor in human evolution, has a positive influence on many human physiological processes and that insufficient sun exposure has become a real public health problem [117]. The positive aspects of moderate UVR exposure on numerous areas of human physiology relate not only to the UV-mediated synthesis of the vitamin D required throughout the organism, but also, as has recently been recognized, to the release of NO by photodecomposition of cutaneous NO precursors, such as the photolabile NO derivatives nitrite and RSNO [79,118,119]. In the current literature, a positive effect is attributed to both UVR-dependent factors on a wide variety of physiological processes, with the positive effect on the cardiovascular system being particularly emphasized [118]. Against this background, it makes perfect sense to point out that a system of increased cutaneous NOD concentration, either through food or exogenous application, as well as moderate UVR exposure, represents a cardiovascular supportive measure [120]. 

Furthermore, although epidemiological, mechanistic, and study data provide solid evidence that sunlight is a risk factor for skin cancer, the prevalence of cardiovascular and cerebrovascular death is about 100 times higher than that of skin cancer. Interventions that result in small changes in the incidence of cardiovascular disease therefore have even greater public health benefits than large changes in skin cancer. Epidemiological and mechanistic data now suggest that sunlight has cardiovascular benefits. A priority of photobiology research should now be the development of advice that balances the established carcinogenic effects of ultraviolet radiation with the possible or probable benefits of the same UV radiation on cardiovascular health and all-cause mortality [118,121].

## 8. Summary and Conclusions

There is a strong line of evidence to show that an increase in NO bioavailability has beneficial effects on health. In this context, bioactive NODs such as nitrite and S-nitrosylated proteins are jointly responsible for NOS-independent NO generation and may act as an NO store or back-up system when oxygen-dependent NO synthesis is compromised. Human skin can store and contain comparatively high concentrations of these NODs. The modulation of these stores, as shown in Figure 5 (e.g., by nitrate-rich diet, NO donor, CAP), and/or the induction of dermal NO release by NOD decomposition (e.g., by UVA/blue light) may represent promising therapeutic strategies against local pathological conditions such as disturbed microcirculation, skin infections, and impaired wound healing, and also against systemic conditions such as hypertension, arteriosclerosis, and diabetes.

However, it seems that many other factors, in particular the levels of antioxidants, have a strong impact on NO release. Moreover, further studies are necessary concerning NOD stores and possible premature skin ageing and/or skin cancerogenesis.

## Figures and Tables

**Figure 1 biomedicines-10-02124-f001:**
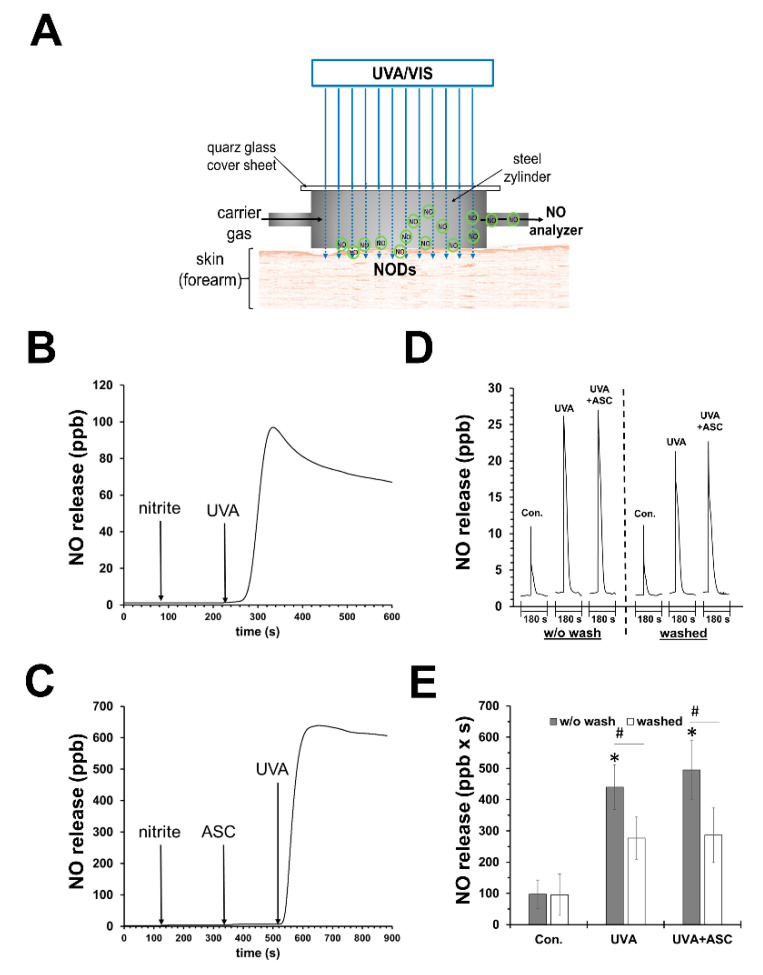
**UVA can induce nitric oxide release by photodecomposition of nitrite.** (**A**) Experimental set-up to measure nitric oxide (NO) emanation from human skin. (**B**) NO release from an UVA-irradiated (70 mW/cm^2^) reaction chamber with saline buffer (PBS; pH 7.4; 20 mL) containing nitrite (10 µM) and (**C**) in addition sodium ascorbate (1 mM). There was not any significant release of NO without nitrite (not shown). (**D**) Representative registration of NO release from skin of one volunteer after 180 s UVA exposure (18 mW/cm^2^) using the experimental set-up pictured in (**A**). Mean values ± SD of integrated NO peaks (*n* = 4) are shown in (**E**); * *p* < 0.05 as compared to the controls and # *p* < 0.05 as compared to respective values obtained from unwashed arms.

**Figure 2 biomedicines-10-02124-f002:**
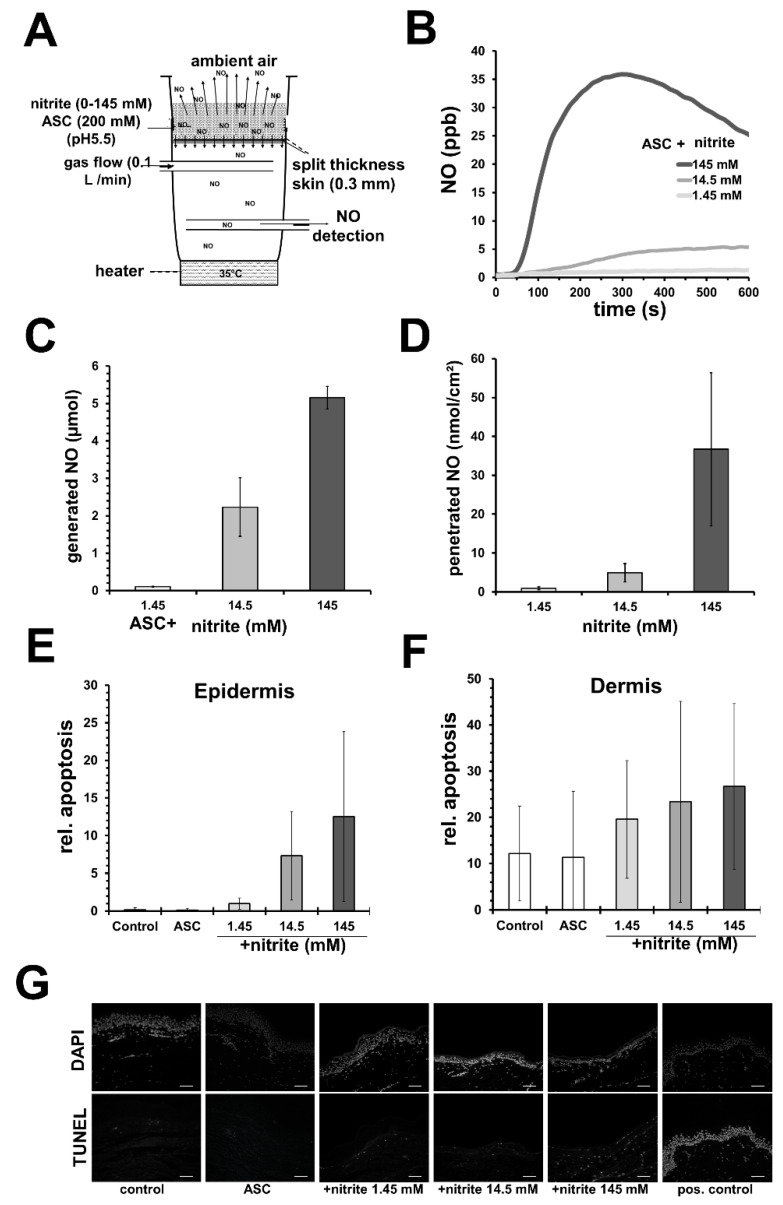
**Nitric oxide**–**releasing solutions can induce apoptosis in human skin**. (**A**) Experimental set-up to measure nitric oxide (NO) penetration through human split skin. (**B**) Exemplary registration of measured NO below skin with sodium ascorbate (ASC; 100 mM)-containing acetate buffer (pH 5.5) and nitrite in concentration as indicated. (**C**) Shows the amount of generated NO from 1 mL of the nitrite solution (described in (**B**) in 600 s). (**D**) By integration the amounts of penetrated NO were calculated. Shown are the mean values ± SD of 3 independent experiments using different skin specimens. (**E**) Different nitrite/ascorbate buffers were applied on vital human skin samples and apoptosis was detected 24 h later by TUNEL assay. Shown are the mean values ± SD (*n* = 3) of relative apoptosis, the counted number of TUNEL-positive cells in relation to the total number (DAPI-stained) of cells found in the epidermis or (**F**) dermis. (**G**) Exemplary fluorescence microscope images obtained from skin samples treated as indicated (white bar = 100 µm).

**Figure 3 biomedicines-10-02124-f003:**
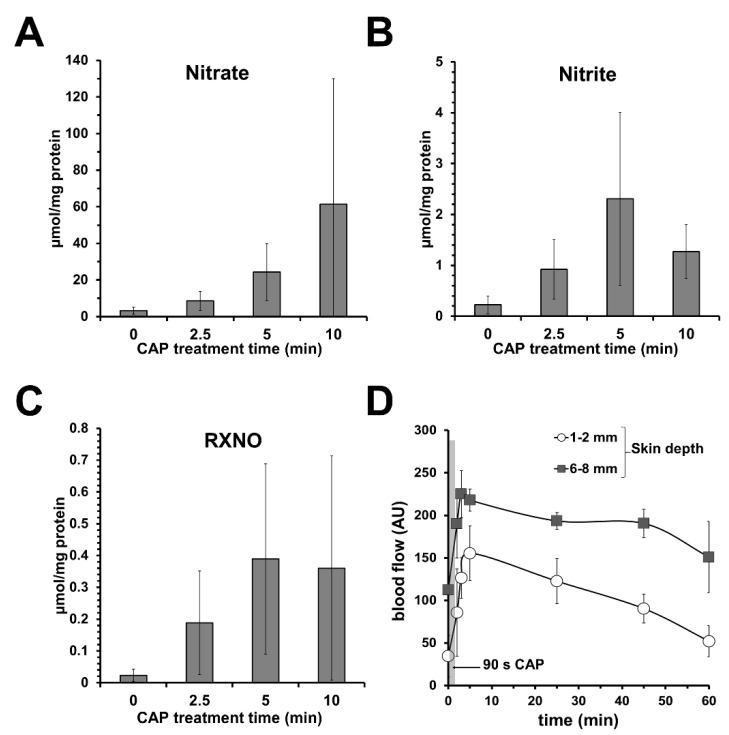
**Topical application of cold atmospheric plasma (CAP) increases dermal microcirculation and the amount of nitric oxide derivates (NODs) in skin.** Using a dielectric barrier device as CAP source, topical CAP treatment of human skin samples led to an increase in (**A**) nitrate, (**B**) nitrite, and (**C**) nitrosylated compounds (RXNOs) found in the respective skin homogenizates. Shown are the mean values ± SD of 6 independent experiments (treated skin surface 0.64 cm^2^). (**D**) Hairless areas of forearm were treated with CAP (90 s) and blood flow was measured in different skin depths as indicated by a microlight guide spectrophotometer (O2C) device. Given are blood flow mean values ± SD of volunteers (*n* = 4) showing a CAP-induced increase in dermal blood flow in the treated area.

**Figure 4 biomedicines-10-02124-f004:**
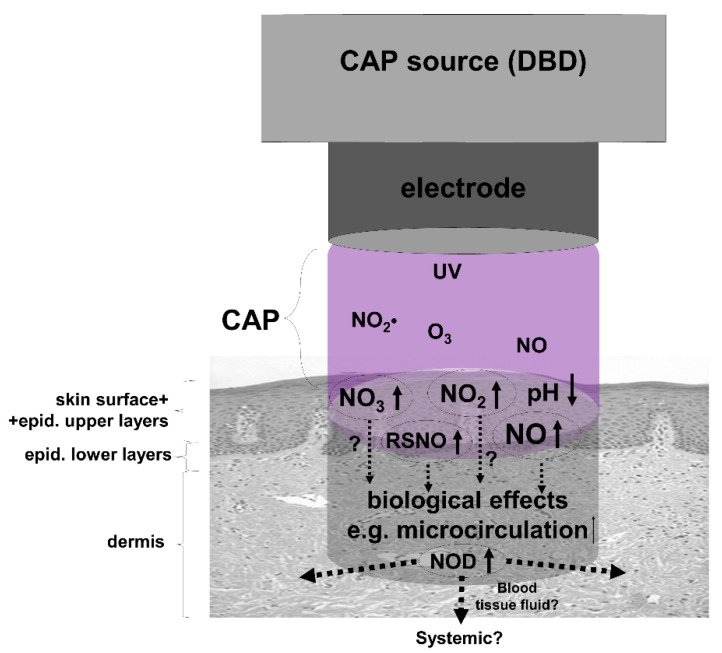
**The use of cold atmospheric plasma (CAP) to enrich skin with nitric oxide derivates (NOD).** Shown is a simplified assumption of possible reactions of CAP during treatment of skin. A dielectric barrier discharge (DBD) device as CAP source operated under ambient air generates, i.e., water-soluble nitrogen dioxide (NO_2_•), which in turn hydrolyzes to nitric acid (HNO_3_) and nitrous acid (HNO_2_) on skin leading to an acidification of the skin surface. Thus, the skin surface, sweat, and the upper skin layers can be loaded with the anions nitrate (NO_3_) and nitrite (NO_2_), which may partially decompose under the induced acidic conditions and UV irradiation generated in CAP to nitric oxide (NO) that can penetrate deeper through the epidermal barrier and/or leading to the formation of S-nitrosylated proteins (RSNO), both capable of increasing local dermal microcirculation.

**Figure 5 biomedicines-10-02124-f005:**
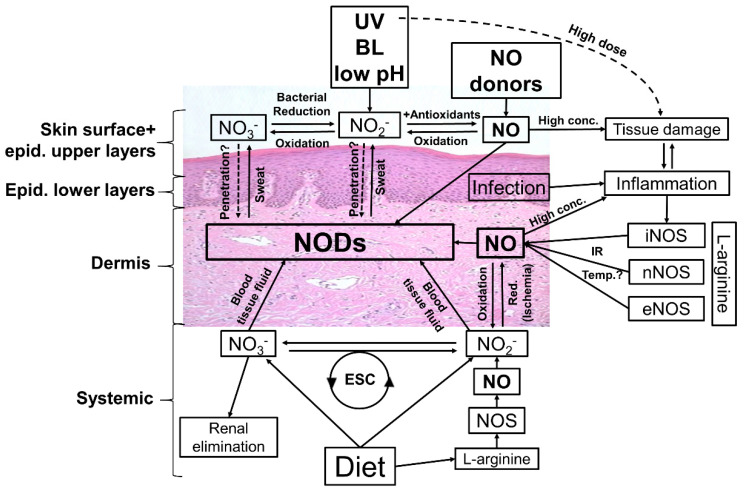
**Possible ways to modulate the nitric oxide derivates (NODs) in the human skin****.** Targeting the skin there are many ways to affect the amounts and composition of NODs in skin tissues. Topical application of nitric oxide (NO) donors and nitrate (NO_3_), nitrite (NO_2_) +/− acidification can increase the NOD concentrations in skin layers and eventually in underlying tissues and circulation. Irradiation with UV/UVA, or blue light (BL)-photolabile NODs found in skin, in particular nitrite, generates NO, which in turn penetrates deeper in the skin and may directly enter the circulatory system and/or react to bioactive NODs, exerting vasodilatory effects and lowering the blood pressure. By tissue damage, infection, and inflammation, the inducible NO synthase (iNOS) is activated, producing high amounts of NO via reduction of L-arginine. The generated NO reacts in the skin with biomolecules (e.g., proteins) or is oxidized to nitrite/nitrate, leading to an increase in NOD content in the skin. It seems possible to enhance the activity of iNOS and also the constitutively expressed eNOS and/or nNOS via higher temperature/infrared radiation (IR). By oral intake of nitrate/nitrite it is possible to increase, with help of the enterosalivary circle (ESC), the general amount of NODs in the human body and theoretically also the NODs in the skin. A major part of nitrate is excreted via the urine, which opens up further possibilities for influencing blood and therefore tissue concentrations of nitrate and perhaps further NODs.

## Data Availability

The data that support the findings of this study are available from the corresponding author upon reasonable request.

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
