# Peer review of "Enhancement of Nitric Oxide Bioavailability by Modulation of Cutaneous Nitric Oxide Stores"

_biomedicines, 2022, doi:10.3390/biomedicines10092124_

Round 1

Reviewer 1 Report

The authors summarize in their review current findings about nitric oxide/NOS and NODs in regard to skin. 

Some comments:

1.The authors should check if the Figures are always cited in the text at the corresponding position. 

2.Figures 1, 2, and 3 display results of the authors that were done and described in other manuscripts? Or is it new data that is incorporated here in this review? It might be unusual to display new findings in a review article rather than in an original article. 

Author Response

Author's Reply to the Review Report

Reviewer #1

Comment 1: The authors should check if the Figures are always cited in the text at the corresponding position. 

Author's Reply: We took up this point when revising the manuscript and made corresponding improvements in a few places.

Comment 2: Figures 1, 2, and 3 display results of the authors that were done and described in other manuscripts? Or is it new data that is incorporated here in this review? It might be unusual to display new findings in a review article rather than in an original article. 

Author's Reply: The data shown are previously published results that are presented here in a graphically modified form.

Corresponding corrections and changes can be tracked in the marked version of the manuscript.

Reviewer 2 Report

Manuscript Number: biomedicines-1848426

Enhancement of nitric oxide bioavailability by modulation of cutaneous nitric oxide stores.

This is an interesting paper, I´ve enjoyed reading and reviewing it. The manuscript is fairly well written, although there are a fair number of grammatical errors throughout. I agree that it would be an important topic to cover and that a review of this topic would be very helpful in condensing all of that information and getting it in one place.

Author Response

Author's Reply to the Review Report

Reviewer #2

Comment: This is an interesting paper, I´ve enjoyed reading and reviewing it. The manuscript is fairly well written, although there are a fair number of grammatical errors throughout. I agree that it would be an important topic to cover and that a review of this topic would be very helpful in condensing all of that information and getting it in one place.

Author's Reply: We have revised the manuscript with special regard to correct English grammar and diction.

Corresponding corrections and changes can be tracked in the marked version of the manuscript.

Reviewer 3 Report

The authors provide a very long and detailed review on many aspects of nitric oxide. The ultimate goal is to suggest that skin may be a useful reserve for providing NO systemically, potentially to treat medical problems. The review is essentially theoretic but the use of skin NO for systemic treatment is not proven clinically. Since treatment with nitrates/nitrites (such as nitroglycerin) has been in medicine for many decades, it is not clear why there is a need to utilize skin NO for diseases. Even more concerning is the concept of using ultraviolet light to induce NO production. UV light is clearly toxic to skin - leading to burns, and increased skin cancer risk. The authors then go on to describe the potential for treating patients with cold atmospheric plasma to induce NO production, but this treatment seems to be even more radical. 

So this theoretic and extensive review seems suggest an impractical treatment that is really not necessary. 

The paper is very long and could be reduced significantly.

Author Response

Author's Reply to the Review Report

Reviewer #3

Comment 1: The review is essentially theoretic but the use of skin NO for systemic treatment is not proven clinically. Since treatment with nitrates/nitrites (such as nitroglycerin) has been in medicine for many decades, it is not clear why there is a need to utilize skin NO for diseases.

Author's Reply: Organic nitrates, such as the aforementioned nitroglycerin, are used exclusively in the context of emergency medical interventions in order to counteract an acute ischemic condition in the affected tissue (mostly heart, angina pectoris) by dilating blood vessels.

There are numerous clinical and clinical-experimental studies that show that nitrite supplementation, administered systemically or ingested as part of the diet, can have a positive effect on the cardiovascular system in particular in the form of increased NO release. In this context, it is not only plausible, it has also been shown experimentally that the NO derivatives present in the skin represent a physiologically accessible store for NO and can supply the organism with NO through translocation from the skin. The nitrite-decompository property of UV radiation or blue light can amplify and specifically control this translocation.

Comment 2: Even more concerning is the concept of using ultraviolet light to induce NO production. UV light is clearly toxic to skin - leading to burns, and increased skin cancer risk.

Author's Reply: We completely agree with the reviewer's assessment of the potential danger of UV radiation. UV radiation can be a harmful and carcinogenic environmental medium and frequent UV exposure with higher radiation doses that can lead to erythema or even sunburn should be avoided at all costs. In order to avoid any misunderstandings, we now mention this fact explicitly in chapter 6.4. UV/VIS modulation of dermal NOD.

However, it is also undisputed that sunlight, as the most important physical environmental factor in human evolution, has a positive influence on many human physiological processes. The positive aspects of moderate UVR exposure on numerous areas of human physiology relate not only to the UV-mediated synthesis of the vitamin D required throughout the organism, but also, as has recently been recognized, to the release of NO by photodecomposition of cutaneous NO precursors, like the photolabile NO derivatives nitrite and RSNO. In the current literature, both UVR-dependent factors are attributed a positive effect on a wide variety of physiological processes, with the positive effect on the cardiovascular system being particularly emphasized. Against this background, it makes perfect sense to point out that a system of increased cutaneous NOD concentration, either through food or exogenous application, and moderate UVR exposure represents a cardiovascular supportive measure.

Furthermore, although epidemiological, mechanistic and study data provide solid evidence that sunlight is a risk factor for skin cancer, however, the prevalence of cardiovascular and cerebrovascular death is about 100 times higher than that of skin cancer. Interventions that result in small changes in the incidence of cardiovascular disease therefore have even greater public health benefits than large changes in skin cancer. Epidemiological and mechanistic data now suggest that sunlight has cardiovascular benefits. A priority of photobiology research should now be the development of advice that balances the established carcinogenic effects of ultraviolet radiation with the possible or probable benefits of the same UV radiation on cardiovascular health and all-cause mortality. We also mention and quote this aspect in Chapter 6.4.

Comment 3: The authors then go on to describe the potential for treating patients with cold atmospheric plasma to induce NO production, but this treatment seems to be even more radical. 

Author's Reply: Even if the use of cold plasmas to enrich the skin with NO appears a bit bizarre and exotic, this measure represents common clinical practice since many years, e.g. in the treatment of wounds. There are numerous studies that have described the antibacterial and wound-healing supporting effects of cold atmospheric plasma. The positive influence of NO-containing plasmas was particularly emphasized when it comes to promoting the healing of chronic and poorly healing wounds. The "charging" of the skin with an NO-releasing system (NOD incorporation plus acidic pH) using plasma technology, which we mention in the manuscript, represents an experiment-based postulate of the mechanism of the known positive effect of plasma treatment with a DBD plasma source. In intact skin ex vivo and in the epiderm tissue model we could not find plasma-induced toxicity even after longer treatment (up to 60 min). Thus, CAP treatment seems not so radical.

Comment 4: So this theoretic and extensive review seems suggest an impractical treatment that is really not necessary. The paper is very long and could be reduced significantly.

Author's Reply: We think it's a pity that with the delivered version we don't really convince the reviewer of the far-reaching role of NO and its derivatives in human physiology, in particular the role of the skin as a collecting basin for NOS-generated and ingested NODs for the cardiovascular system.  Furthermore, we cannot share the opinion of the reviewer that the application suggestions we have described are impractical and unnecessary. The current literature and the development of other NO-based and light- or plasma-induced therapy options represent the opposite opinion.

The manuscript actually turned out to be longer than we originally intended. However, we believe that in order to present the "story" in an understandable and complete way, this length is justified. Due to the reviewer's criticism, we were able to determine, when reviewing the manuscript, that Chapter 6.5. Targeting the activity of NO synthases and Chapter 6.6. Targeting the renal clearance of nitrate did not fit the context of the review as much due to the impractical clinical implementation and we therefore removed the chapters from the manuscript

If the reviewer continues to see a radical abridgement of the manuscript as a prerequisite for possible acceptance of the manuscript, we would do so with a heavy heart, so as not to deprive the reader of what we believe to be an important new aspect of human physiology.

All corrections and changes made can be tracked in the marked version of the manuscript.

Round 2

Reviewer 3 Report

I do not feel that the authors have addressed my concerns and I have not changed my view of the paper. 

Author Response

Dear Reviewer,

Sorry, we don't know how to respond. We tried to refute each of the criticisms  expressed in the first review process meticulously and with arguments. However, the new commentary, these points were ignored at all, and the comments expresse a blanket judgment to which we cannot react adequately.

S